

# Decrease of total electron content during the 9 March 2016 total solar eclipse observed at low latitude stations, Indonesia

Wahyu Srigutomo[1], Alamta Singarimbun[1], Winda Meutia[1], I Gede Putu Fadjar Soerya Djaja[1], Buldan Muslim[2] and Prayitno Abadi[2]

[1]Physics of Earth and Complex System, Physics Dept., Fac. of Mathematics and Natural Sciences, InstitutTeknologi Bandung, Jl. Ganesa 10, Bandung 40132, Indonesia
[2]Indonesian National Institute of Aeronautics and Space (LAPAN), Jl. Dr. Djunjunan No.133, Bandung 40173, Indonesia

*Correspondence to*: Wahyu Srigutomo (wahyu@fi.itb.ac.id)

**Abstract.** The total solar eclipse on 9 March 2016 was a rare phenomenon that could be observed in 12 provinces in Indonesia.
The decline in solar radiation to the earth during a total solar eclipse affects the amount of electron content (TEC) in the ionosphere. The ionospheric dynamics during the eclipse above Indonesia have been studied using data from 40 GPS stations distributed throughout the archipelago. It was observed that TEC decrease occurred over Indonesia during the occurrence of the total eclipse. This TEC decrease did not instigate ionoshperic scintillation. Moreover, the relationship between eclipse magnitude and TEC decrease throughout three GPS stations was analyzed using PRN 24 and PRN 12 codes. Data analysis
from each station reveals that the time required by the TEC to achieve maximum reduction since the initial contact of the eclipse is faster than the recovery time. The maximum TEC reduction came about several minutes after the maximum obscuration indicating that the recombination process was still ongoing even though the peak of the eclipse had happened. The magnitude of this decline is positively correlated with the geographical location of the stations and the relative satellite trajectory with respect to the total solar eclipse trajectory. The amount of TEC reduction is proportional to the magnitude of
the eclipse which is directly related to the photoionization process. Because Indonesia is located in a low latitude magnetic equator region, the dynamics of the ionosphere above it is more complex due to the fountain effect. During the solar eclipse, the fountain effect declines disturbing the plasma transport from the magnetic equator to low latitude regions.

## 1  Introduction

Atmospheric dynamics occur due to internal factors of the solar activities such as coronal mass ejection (CME), solar storms,
sunspots, solar prominence, corona holes, and flares (Ruohoniemi et al., 2001; Kumar and Singh, 2012: Meza et al., 2009). There are also external factors such as solar eclipse, a situation when the moon passes between the earth and the sun, yielding an occultation of the solar radiation from reaching the earth (Sharma et al., 2010; Hoque et al., 2016; Momani et al., 2010). During the solar eclipse, a reduction in the flux of solar radiation in the ionosphere causes several conditions such as a decrease in electron density in the E layer and F1 layer due to the dynamics of plasma transport and chemical reactions (Ding et al. 2010; Sharma et al., 2010; Galav et al., 2010; Khumar and Sigh, 2012), changes in the height of the ionosphere layer boundary



(Jayakrishan et al., 2013) and several phenomena related to the generation and propagation of gravitational waves (Farges et al., 2003).

Ionosphere dynamics during the occurrences of solar eclipses can be analyzed based on the total electron content (TEC). TEC is the total number of electrons along the path between the satellite and the receiving station or receiver. The TEC unit (TECU) is electron per square meter where $10^{16}$ electrons m$^{-2}$ = 1 TECU (Adewale, 2011). Data retrieval in this study was carried out using (Global Positioning Systems (GPS) receivers since they are able to cover a wide ionosphere area and the data have a high measurement accuracy (Baran et al., 2016; Tsai et al., 1999; Le et al., 2009; Afraimonivch et al., 2002). Based on several previous studies during the solar eclipse, the decrease in fluctuations in TEC at high, middle and low latitudes has a different pattern. Observation of the ionospheric response in the equatorial anomalies during the total solar eclipse on October 4, 1995 and March 9, 1997 showed that the TEC response at low latitudes was strongly influenced by the fountain effect that causes plasma transport from the magnetic equator to the crest area to be decreasing (Tsai and Liu, 1999). The decreasing of TEC can reach 30 - 40%.

The TEC complexity during the occurrence of a total solar eclipse over low latitude areas such as Indonesia is important in providing information to the users of communication system, navigation and positioning systems. This study presents the effects of solar eclipse on TEC behavior both temporally and spatially during the 9 March 2016 total solar eclipse over Indonesia. Data were taken simultaneously using 40 GPS receiver stations spread all across Indonesia.

## 2  Data and data processing

During the occurrence of a solar eclipse, the diameter of the sun covered by the moon will determine the relative magnitude of the eclipse in various regions. Based on the Indonesian Meteorological, Climatological and Geophysical Agency (BMKG), the totality of the solar eclipse on 9 March 2016 passes through several regions in Indonesia with magnitudes >1 (BMKG, 2016). The path of solar eclipse totality and the locations of 40 GPS receiver stations used for TEC recording in this study is shown in Fig. 1. Data of solar eclipse magnitude and obscuration were obtained from the Solar Eclipse Map website managed by NASA (https://eclipse.gsfc.nasa.gov). All the GPS receiver stations are of Continuously Operating Reference Station (CORS) managed by the Indonesian Geospatial Information Agency (BIG). All the receivers can record dual frequency signals operated at L1 (1575.42 MHz) and L2 (1227.60 MHz). The GPS receiver stations receive signals transmitted by 24 satellites orbiting at an altitude of 20,000 km. To get to the earth, these signals must first pass through the ionosphere. The integration of the number of electrons in the ionosphere is proportional to the delay between L1 and L2. The data obtained from each receiver were still in the Receiver Independent Exchange Format (RINEX) and need to be processed into TEC format. The format conversion was carried out using GPS_GOPI software (http://seemala.blogspot.com/2017/09/gps-tec-program-ver-295.html) to obtain the appropriate TEC parameters.

The calculation of the slant TEC (STEC) for an arbitrary path can be carried out using the following equation (Ya'acob et al., 2010; Arikan et al., 2002; Patel et al., 2017)



$$STEC = \int_0^s N(s)\,ds = \left[\frac{f_2^2}{f_1^2 - f_2^2}\right]\frac{2f_1^2}{K}\Delta P_{1,2} \qquad (1)$$

where $N$ is the electron density and $s$ is the length of the wave path from the satellite to the GPS receiver on the earth's surface. $\Delta P_{1,2}$ is the difference between time delays of L1 and L2 signals, $K = 40.3$ m$^3$s$^{-1}$, $f_1$ and $f_2$ are the frequencies of L1 and L2 signals, respectively. For enabling TEC to be mapped vertically across the earth's surface, vertical TEC (VTEC) definition is

often used and expressed as (Manucci et al., 1993; Langley et al., 2002; Arikan et al., 2002)

$$VTEC = \frac{STEC - [b_s + b_r]}{S(E)}, \qquad (2)$$

where $b_s$ is the satellite bias associated with the GPS satellite differential delay and respectively, $b_r$ is the receiver bias associated with the GPS receiver differential delay. $S(E)$ is the slant factor given by

$$S(E) = \frac{1}{\cos(z')} = \left[1 - \left(\frac{R_e + \cos(z)}{R_e + h}\right)^2\right]^{-1/2}. \qquad (3)$$

In the above equation, $R_e$ is the average of earth radius (6371 km), $z$ is the elevation angle, $z' = (90° - z)$ and $h$ is the altitude of the ionospheric pierce point (IPP) taken to be 350 km since it is valid for low latitude zone with elevation angle > 50° (Rao et al., 2006).

To ensure that the TEC data used are not influenced by other factors, an analysis of geomagnetic storm events was also carried out. Based on World Data Center (WDC) for Geomagnetism (http://wdc.kugi.kyotou.ac.jp/dst_realtime/

201603/index.html), Husin (2016) reported that there was a moderate geomagnetic storm on 6 March 2016. However, the geomagnetic storm influence to TEC only occurred on 7 March and had no effect on the day after. Therefore, the ionospheric condition on 8 March is regarded as the normal condition to be compared with the ionospheric condition on 9 which is highly affected by the total solar eclipse.

In this study, we employed VTEC data to disclose the effects of solar eclipse on the low-latitude ionosphere over

Indonesia. The analyzed VTEC distribution are based on data recorded on 8 and 9 March 2016 from 5:00:00 WIB or Local Time (LT = UTC + 7:00:00) until 11:00:00 LT. The data were filtered to include only signals from all satellites having elevation angle > 10°. The filtering was carried out to avoid disturbances such as multipath, noises caused by high-rise buildings or tall trees, or other causes that are not derived from the effects of the ionosphere. To investigate the relationship between the decrease in VTEC and the magnitude of the solar eclipse at a certain coordinate, data were collected from 3 stations out of all

stations. Geographically each station is located at different coordinates so they will sense different eclipse magnitudes. From each station, two pseudorandom noise (PRN) GPS satellites were selected, the data of which will be analysed further based on their visibility to the time of the eclipse event ensuring the data recording was executed during the eclipse occurrence.



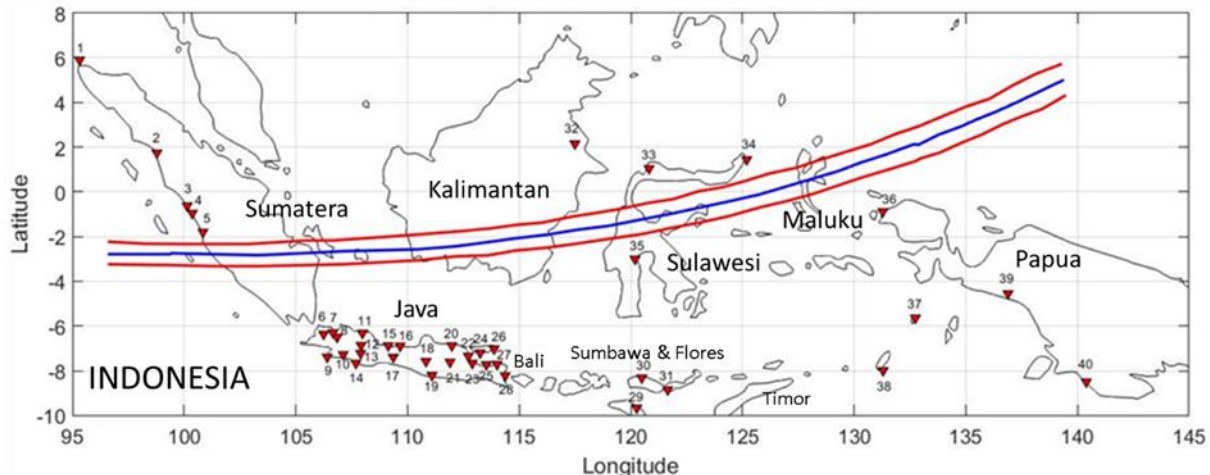

**Fig. 1.** Totality path of the 9[th] March 2016 solar eclipse over the Indonesian archipelago at low latitudes. The blue line denotes the central position of the moon and the two red lines denote the northern and southern limits of area whose eclipse magnitudes equal to 1. The red triangles represents the locations of GPS receiver stations. Note that the stations are more densely distributed in Java Island.

## 3 Results and discussion

Based on the BMKG report, the earliest time of the eclipse took place in Kotaagung, Lampung, Sumatera island at 06:19:41 WIB (local time in western part of Indonesia, UTC + 7 hours) and the final contact time of the eclipse occurred in Jayapura, Papua island at 11:48:46.6 WIT (local time in eastern part of Indonesia) or 9:48:46.6 WIB. Fig. 2 (a), (c), (e), (g), (i) and (k) depict the VTEC distribution maps over Indonesia on 8 March 2016 at one hour interval started from 5:00:00 and ended at 10:59:59. These maps are regarded as the background maps portraying normal conditions, whereas Fig. 2 (b), (d), (f), (h), (j) and (l) depict VTEC distribution maps on 9 March 2016 during the occurrence of the eclipse at the same time intervals of those maps from the 8 March 2016.

VTEC distribution maps shown in Figure 2 show that there is an increase in the TEC values within the interval of 5:00:00 - 10:59:59 WIB on 8 and 9 March 2016. For both days, higher concentrations of TEC were found in central and eastern part of Indonesia. This is due to the higher sun position in those areas so that the reception of solar radiation and the photoionization process became more intense. By comparing the VTEC values between the 8 and 9 March 2016 at the same hours, it is found that the TEC on the 9 March experienced a decrease compared to that on 8 March. The decrease in the TEC was due to the effect of the total solar eclipse occurred on 9 March 2016.

Figure 2 (a) and (b) show VTEC distribution maps at 5:00:00-05:59:59 WIB on 8 and 9 March 2016, respectively. At this time interval on 9 March, the initial contact of the solar eclipse had not yet occurred, consequently, the decline in the TEC was not obviously visible. Whereas, the decrease in TEC concentration at time interval 6:00:00-6:59:59 WIB on 9 March is



observed particularly in the eastern part of Indonesia as can be seen in Figure 2 (c) and (d). Although not so significant, the decrease in the TEC has a positive correlation with the initial contact of the eclipse which started at 06:19:41 WIB and ended at 06:53:41 WIB. Figure 2 (e), (f), (g) and (h) indicate that the decline in the TEC was increasingly clear because it relates to the peak time of the eclipse, which started at 07:19:49 WIB in the western part of Indonesia and ended at 08:17:41 WIB in the eastern part of Indonesia. The decline in the TEC did not only occur on the path of totality of the eclipse, but almost in all coordinates over Indonesia. Interestingly, as can be seen in Figure 2 (i) and (j), a larger decrease in the concentration of TEC in eastern region of Indonesia occurred on 9 March regarding to that of 8 March, but in reverse, an increase in TEC was observed in the western regions of Indonesia. This can be explained by referring to the final contact time of the eclipse which started at 08:24:46 WIB and ended at 09:48:46.6 WIB where the solar eclipse on its path of totality has reached the eastern region of Indonesia causing the effects of the eclipse position on the decline in TEC to be strongly perceived in the eastern region.

Figure 3(a) depicts the average TEC values from all GPS stations over Indonesia on 8 March 2016 designated as the background or normal TEC condition, and the TEC values on 9 March at hours of eclipse occurrence. The maximum decrease in TEC is about 5.6 TECU as can be seen from Figure 3 (b). In addition, observations of the average S4 index were also performed to see the occurrence of scintillation during solar eclipse. Abadi et al. (2014) classified S4 values into 3 categories: weak (0.2 <S4 <0.5), moderate (0.5 < S4 <0.8), and strong (S4> 0.8). In this study, the maximum S4 value was found to be 0.114 during the total solar eclipse. This value is still below the weak category making it plausible to confirm that interference of natural disturbance in form of ionospheric scintillation with the satellite signals did not occur during the total solar eclipse. Profile of S4 index on 9 March 2016 is depicted in Figure 3 (c).

To investigate the behavior of TEC related to the magnitude of the eclipse during the total solar eclipse on 9 March 2016, an analysis of two PRN numbers namely PRN 24 and PRN 12 from three GPS receiver stations was carried out. Those three stations are CNDE, CREO and CSMN (Figure 4). CSMN station is located closer to the totality path of the solar eclipse followed by CREO and CNDE stations. The location of GPS stations and trajectory of each PRN observed in this study are shown in Figure 4.

**Table 1.** Local condition at the time of total solar eclipse at each stations

| Station | Longitude (E) | Latitude (S) | Time of contact (WIB) | | | Magnitude | Obscuration |
|---------|---------------|--------------|-----------|-----------|-----------|-----------|-------------|
| | | | First contact | Max. contact | Last contact | | |
| CNDE | 121.6545 | 8.8595 | 06:26 | 07:35 | 08:53 | 0.773 | 72.14% |
| CREO | 120.363 | 8.4743 | 06:25 | 07:33 | 07:51 | 0.792 | 74.54 % |
| CSMN | 114.0596 | 7.7218 | 06:21 | 07:26 | 08:41 | 0.842 | 80.71 % |



**Fig. 2.** (a), (c), (e), (g), (i) and (k) are VTEC distribution maps on 8 March 2018 at one hour interval started from 05:00:00 WIB (UTC + 7 hours). (b), (d), (f), (h), (j) and (l) are the VTEC distribution maps at the same time interval on 9 March 2018 during the occurrence of the total solar eclipse.





The solar eclipse at CNDE began at 6:26 WIB and ended at 08:53 WIB, whereas at CREO it started at 06:25 WIB and ended at 08:51 WIB. Meanwhile at the CSMN the eclipse started at 06:21 WIB and ended at 08:41 WIB. The difference in the geographical position of the station causes a difference in the time of the eclipse, the magnitude of the obscuration and the

magnitude of the eclipse in each station. Parameters of the total solar eclipse observed in all three stations are listed in the Table 1.

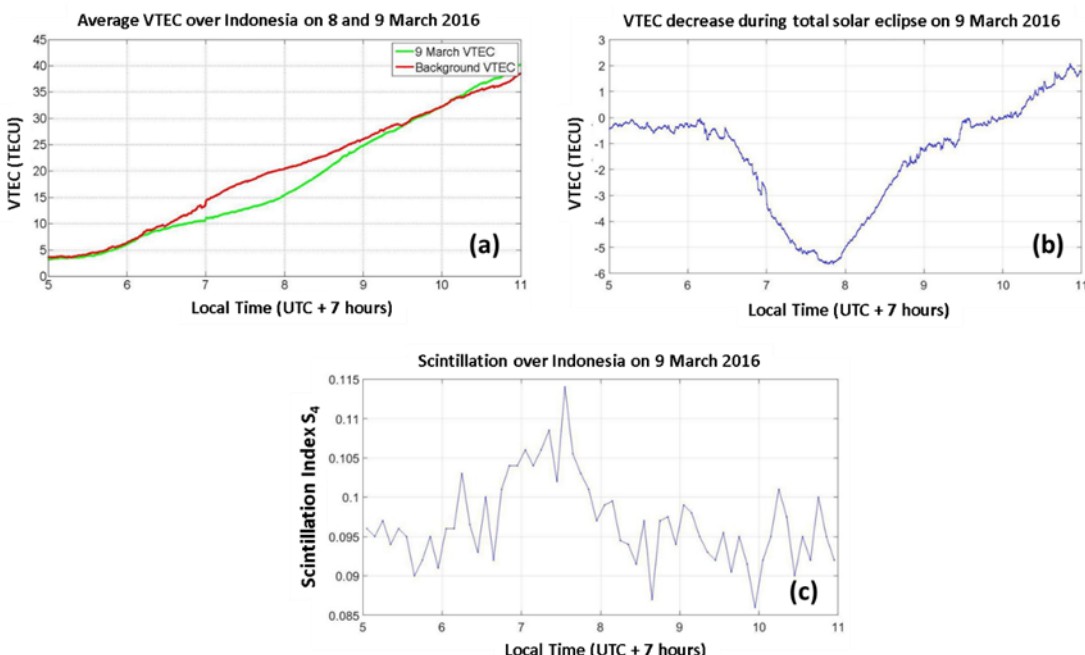

**Fig. 3.** (a) Average VTEC over Indonesia on normal condition (8 March 2016) and during the total solar eclipse (9 March

2016). (b) The decrease of TEC value during the occurrence of total solar eclipse. (c) Average scintillation index S4during the solar eclipse.

The variability and decrease percentage of the TEC in PRN 24 and 12 during the solar eclipse on 9 March and during the normal condition one day earlier are plotted in Figure 5 (a), (b) and (c) for CNDE, CREO and CSMN stations, respectively.

The time interval of measurements covers start time, maximum time when the solar eclipse reached its totality and end time as denoted by vertical S, M and E lines, respectively in the figure.

Figure 5 depicts the TEC value on 8 and 9 March 2016 and TEC decrease during the solar eclipse relative for PRN 24 and 12 at CNDE, CREO and CSMN stations. The TEC decrease during the same recorded time for CNDE station measured on PRN 24 and PRN 12 is about 22.87% for PRN 24 and 29% for PRN 12. The TEC decrease on PRN 24 and PRN 12 for

CREO station is respectively about 23.78% and 45.73%. Whereas the TEC decrease on PRN 24 and PRN 12 for CSMN station



is 69.51% and 70.68%. From the above results, we see that the decline in TEC is directly proportional to the magnitude and the obscuration of the solar eclipse obscuration which indicates the amount of radiation received in the area. The closer the PRN path is to the total eclipse path, the greater the decrease in the TEC is. This study also confirms previous research conducted by Sharma et al. (2010) discussing the ionospheric response when an annular solar eclipse occurred in India. The

5    magnitude of TEC decrease as the distance between the observation station and the eclipse pathway is farther.

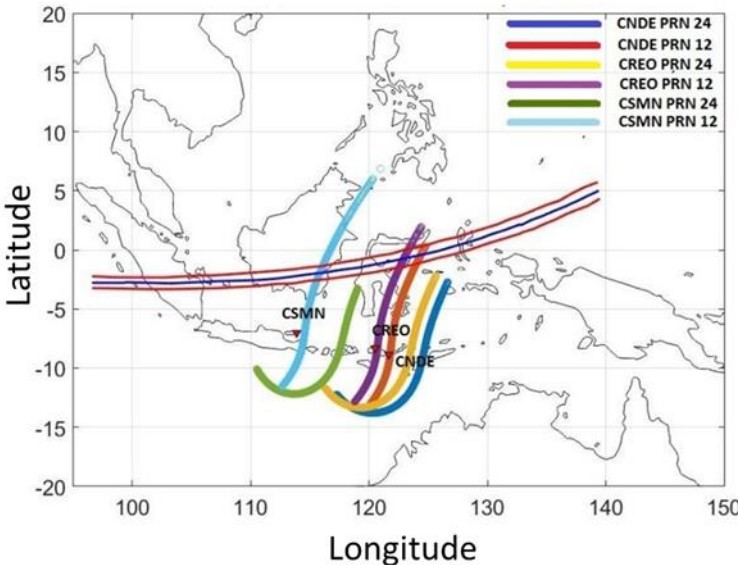

**Fig. 4.** Location of CSMN, CREO and CNDE stations and paths of the satellites labeled by PRN 12 and 24.

10    Previous studies reported that a decrease in TEC can occur several hours and shortly before the initial contact of eclipse. The decline in TEC that occurred almost six to eight hours before the start of the eclipse was reported by Chen et al. (1999). They suspected that this pattern is due to the solar corona obscuration occurring earlier than the optical disk obscuration. Furthermore, the occurrence of initial contact at ionospheric heights always precedes initial contact on the ground. However, a decrease in TEC may also occur simultaneously or shortly before the occurrence of an eclipse (Kumar and Singh, 2011).

15    From Figure 5 (a) the same pattern at CNDE station is also observed where the decrease in TEC started four minutes before the initial contact for PRN 24 and 35 minutes before the initial contact for PRN 12. Meanwhile from Figure 5 (b), CREO station detected a decrease in TEC starting four minutes before the initial contact for PRN 24 and 30 minutes before the initial contact at PRN 12. From Figure 5 (c), CSMN station on the other hand, observed a decrease in TEC more than one hour before the first contact.



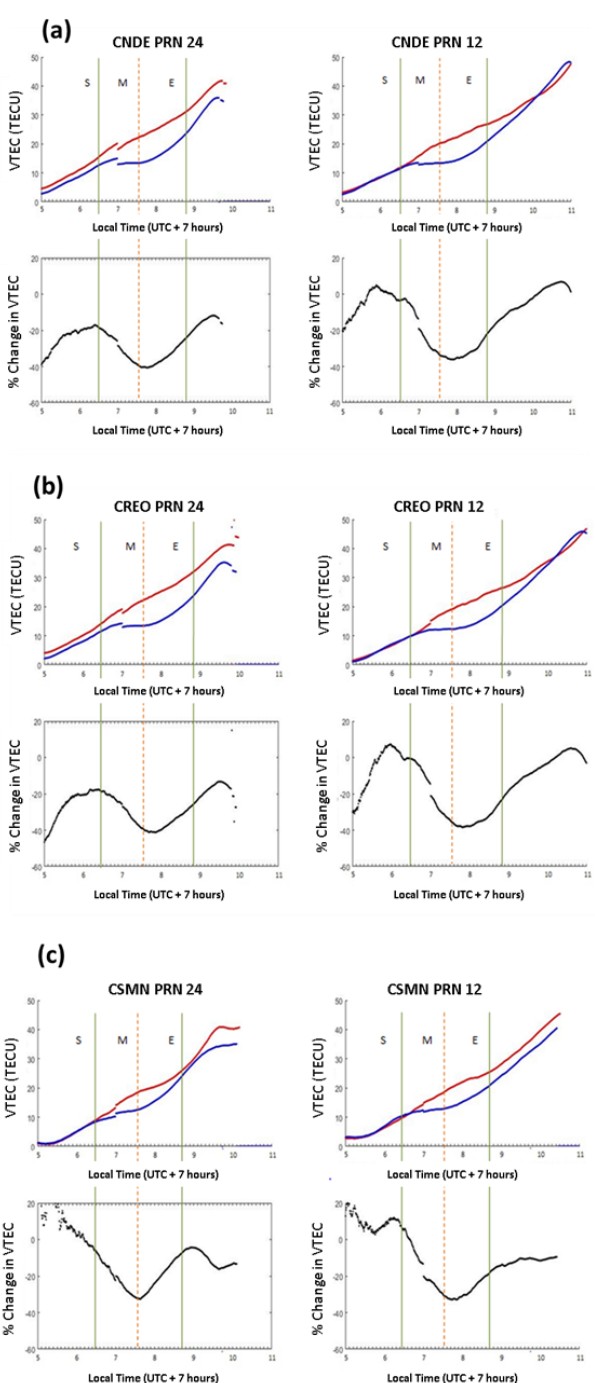

**Fig. 5.** Comparison of TEC variation on 8 March 2016 (red lines) with that of 9 March 2016 (blue lines) and TEC decrease percentage based on PRN 24 and PRN 12 signals at (a) CNDE, (b) CREO and (c) CSMN stations. Vertical lines S, M and E denote the start time, maximum time when the solar eclipse reached its totality and end time of the eclipse, respectively.



It is observed also that the TEC has decreased at a slower rate in a few minutes before and after the initial contact of the eclipse, marked by a gentle slope of TEC decrease with respect to time. However, the TEC decline became faster after those times as it is marked by steeper slopes towards the maximum decline. At CNDE station for PRN 24 the decrease in the TEC at the beginning of the eclipse was about 0.24 TECU/minute and increased to 0.31 TECU/minute. Whereas for PRN 12 at the

same station the initial TEC decrease was 0.16 TECU/minute and increased to 0.65 TECU/minute. Meanwhile at CREO station for PRN 24, the rate of TEC decline at the beginning of the eclipse was 0.21 TECU/minute and increased to 0.43 TECU/minute. At the same station for PRN 12, the TEC decrease at the beginning of the eclipse was 0.23 TEC /minute and followed by 0.64 TECU/minute. The TEC decrease at CSMN station had already begun before 05:00:00 WIB.

Muslims et al. (2018) argued that that the phenomenon of a slower TEC reduction rate during the initial contact is because

the F layer has not contributed actively in the recombination process compared to the E and D layers. A few minutes after the initial contact the TEC decrease become faster because the recombination process has started to occur actively in the F layer. In this study it is also found that the TEC decrease reached a maximum value several minutes after maximum obscuration. This may imply that the recombination process was still ongoing even though maximum obscuration had been reached.

Obscuration during a solar eclipse results in a decrease in the temperature of ions and electrons, but the temperature of

electrons can drop faster than ions (Evans, 1965). Faster electron temperature decreases causes both mobility of electrons and ions to differ so that electrons will experience downward diffusion to the lower layers and will disappear along with the recombination process. This downward diffusion of electrons results in a reduction of electrons in the F2 layer (Yeh et al., 1997; Kumar and Singh, 2011). The above mechanism can be related to the time length before reaching the maximum TEC decrease and the time length when the TEC values return to their normal condition associated with the transportation process

which is slower than the photoionization process (Chen et al., 1999).

The total solar eclipse on 9 March was also a unique phenomenon in addition to its long periodization. It started in the early morning hours and appeared in the low equatorial magnetic latitude which is referred to as the crest of equatorial ionization anomaly (EIA). At the equator and low latitude regions, the northward magnetic field and eastward electric field are responsible for the upward **E** x **B** plasma drift during daylight which is also known as equatorial fountain effect (Kelley et

al., 1979; Liu et al., 2007). At a certain altitudes where the compressive force is greater and the influence of gravity becomes significant, the electrons will move to 20 degrees to the north and south of the equator following the magnetic field line as the guide center. The electrons in the crest, especially over Indonesia, are dominated by transported electrons from the magnetic equator. Chen et al. (1999) explained that the strength of the fountain effect will weaken when the solar eclipse takes place. In this case, the process of electron transport from magnetic equator to low latitude region will also decrease. This causes the

TEC decrease at the low latitude regions will be greater than that of the middle and high latitudes.



## 4 Conclusions

Decrease in TEC during the total solar eclipse occurred on 9 March 2016 over Indonesia is confirmed from this study. The decrease did not affect the occurrence of scintillation as evidenced by the average scintillation value above Indonesia which is still below the value of low category. The rate of decline in TEC is slower during the initial contact of the solar eclipse. This may be attributed to the influence of the recombination process in F layer which is less active than the process in D and E layers. In addition, the maximum decrease in TEC took place several minutes after the maximum obscuration or peak eclipse time. The time needed by the TEC to achieve maximum reduction since the initial contact of the eclipse was faster than the recovery time. Since Indonesia is located at low latitude the TEC decline is also affected by the weakening of the fountain effect during the eclipse.

## Acknowledgement

The authors wish to express their gratitude to the Geospatial Information Agency (BIG) of Indonesia for providing the GPS data. They are also indebted to all members of Modeling and Inversion Laboratory, Physics of Earth and Complex Systems ITB for their kind assistance and permission to use its facilities during the research.

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
