# Peer review of "Decrease of total electron content during the 9 March 2016 total solar eclipse observed at low latitude stations, Indonesia"

_Annales Geophysicae, 2019_

## Referee Comment (RC1) · Anonymous Referee #1 · 30 Jan 2019

Comments on

Manuscript Number: angeo-2019-11 "Decrease of total electron content during the 9 March 2016 total solar eclipse observed at low latitude stations, Indonesia" by Wahyu Srigutomo, Alamta Singarimbun, Winda Meutia, I Gede Putu Fadjar Soerya Djaja, Buldan Muslim and Prayitno Abadi

The solar eclipse is a short-term phenomenon in which many features of change of ionospheric parameters are pressed in a small time interval. In the given paper, the detailed study of behavior of the total electron content ĐćĐŢĐą is executed over Indonesia during a solar eclipse on March, 9th, 2016. It is shown, that in this area behavior of ĐćÐŢÐą possesses certain features. The paper corresponds to topics of this journal and can be recommended to the publication after the account of the following comments.

The main comments 1. Page 2, a line 22: it is desirable to specify, what system of co-ordinates (geographical or geomagnetic) Figure 1 is presented in. The same concerns to Figure 2. 2. Page 4, lines 14-19: a little inconsistent paragraph. On the one hand, in lines 14-17 it is underlined, that during the main period of the eclipse an increase of ĐćÐŢÐą is observed, on the other hand, in lines 17-19 a decrease of ĐćÐŢÐą is marked. It is desirable to clarify. Also, it is necessary to explain, what means a word-combination Âńconcentrations of TECÂż: value, size, color? 3. Page 5, a line 15: Table 1 has appeared earlier than the reference to it. It is desirable to specify number of station from Figure 1.

Some small remarks 1. Page 1, a line 13: to replace ÂńionoshpericÂż. 2. Page 1, a line 16, page 11, a line 7: faster to replace on less. 3. Page 3, a line 20: are to replace on is. 4. Page 8, a line 5: confuses a word ÂńfartherÂż. 5. Page 10, a line 9: can be in a word-combination Âńthat that theÂż there is anything superfluous? 6. Page 11, a line 22: respon to replace on response. 7. Page 11, a line 28: respons to replace on response. 8. Page 11, a line 29: atmos to replace on Atmos. 9. Page 12, a line 10: to move the reference to page 12, a line 31. 10. Page 12, a line 22: new to replace on New. 11. Page 12, a line 29: march to replace on March.

Please also note the supplement to this comment:
https://www.ann-geophys-discuss.net/angeo-2019-11/angeo-2019-11-RC1-supplement.pdf

---

## Author Comment (AC1) · 3 Feb 2019

We are very grateful pleased for the constructive and encouraging reviews provided by first reviewer of our manuscript.

AC is abbreviation for Author Comment

The main comments 1. Page 2, a line 22: it is desirable to specify, what system of co-ordinates (geographical or geomagnetic) Figure 1 is presented in. The same concerns to Figure 2.

AC: Geographic coordinate system is used in this study. We add this information in the

text and also captions of Fig.1 and Fig.2.

Original: The path of solar eclipse totality and the locations of 40 GPS receiver stations used for TEC recording in this study is shown in Fig. 1. Revision: The path of solar eclipse totality and the locations of 40 GPS receiver stations used for TEC recording in this study are mapped using geographic coordinate system in Fig. 1.

Original: Fig. 1. Totality path of the 9th March 2016 solar eclipse over the Indonesian archipelago at low latitudes. Revision: Fig. 1. Totality path of the 9th March 2016 solar eclipse over the Indonesian archipelago at low geographic latitudes.

Original: Figure 2 (a) and (b) show VTEC distribution maps at 5:00:00-05:59:59 WIB on 8 and 9 March 2016, respectively. Revision: Figure 2 (a) and (b) show VTEC distribution maps at 5:00:00-05:59:59 WIB on 8 and 9 March 2016, respectively, using geographic coordinate system.

Original: Fig. 2. (a), (c), (e), (g), (i) and (k) are VTEC distribution maps on 8 March 2018 at one hour interval started from 05:00:00 WIB (UTC + 7 hours). Revision: Fig. 2. (a), (c), (e), (g), (i) and (k) are VTEC distribution maps on 8 March 2018 at one hour interval started from 05:00:00 WIB (UTC + 7 hours) drawn in geographic coordinate system.

2. Page 4, lines 14-19: a little inconsistent paragraph. On the one hand, in lines 14-17 it is underlined, that during the main period of the eclipse an increase of ĐćĐŢĐą is observed, on the other hand, in lines 17-19 a decrease of ĐćĐŢĐą is marked. It is desirable to clarify. Also, it is necessary to explain, what means a word-combination Âńconcentrations of TECÂż: value, size, color?

AC: We want to express that based on VTEC maps in Fig. 2 there are two important points: 1. Generally, both on 8 and 9 March 2016 the TEC values are higher in central and eastern part of Indonesia than those in western part of Indonesia. This feature is due to the higher position of the sun in the central and eastern parts of Indonesia.

2. Comparison between TEC values on 8 March and 9 March confirms that the TEC values on 9 March are lower than those of 8 March. This is due to the solar eclipse effects.

To void the inconsistency we modified several sentences: Original: VTEC distribution maps shown in Figure 2 show that there is an increase in the TEC values within the interval of 5:00:00 15 - 10:59:59 WIB on 8 and 9 March 2016. For both days, higher concentrations of TEC were found in central and eastern part of Indonesia. This is due to the higher sun position in those areas so that the reception of solar radiation and the photoionization process became more intense. By comparing the VTEC values between the 8 and 9 March 2016 at the same hours, it is found that the TEC on the 9 March experienced a decrease compared to that on 8 March. The decrease in the TEC was due to the effect of the total solar eclipse occurred on 9 March 2016.

Revision: VTEC distribution maps shown in Figure 2 show higher values of TEC were found in central and eastern part of Indonesia for both days of 8 and 9 March 2016. This is due to the higher sun position in those areas so that the reception of solar radiation and the photoionization process became more intense. However, by comparing the VTEC values between those of 8 and 9 March 2016 at the same hours, it is found that the TEC values on the 9 March experienced a decrease compared to those on 8 March. The decrease in the TEC values was due to the effect of the total solar eclipse occurred on 9 March 2016.

We also have erased a redundant sentence: Original: The variability and decrease percentage of the TEC in PRN 24 and 12 during the solar eclipse on 9 March and during thenormal condition one day earlier are plotted in Figure 5 (a), (b) and (c) for CNDE, CREO and CSMN stations, respectively. The time interval of measurements covers start time, maximum time when the solar eclipse reached its totality and end timeas denoted by vertical S, M and E lines, respectively in the figure. Figure 5 depicts the TEC value on 8 and 9 March 2016 and TEC decrease during the solar eclipse relative for PRN 24 and 12 at CNDE, CREO and CSMN stations. The TEC decrease

during the same recorded time for CNDE station measured on PRN 24 and PRN 12 is about 22.87% for PRN 24 and 29% for PRN 12.

Revision: The variability and decrease percentage of the TEC in PRN 24 and 12 during the solar eclipse on 9 March and during the normal condition one day earlier are plotted in Figure 5 (a), (b) and (c) for CNDE, CREO and CSMN stations, respectively. The time interval of measurements covers start time, maximum time when the solar eclipse reached its totality and end time as denoted by vertical S, M and E lines, respectively in the figure. The TEC decrease during the same recorded time for CNDE station measured on PRN 24 and PRN 12 is about 22.87% for PRN 24 and 29% for PRN 12.

Also, it is necessary to explain, what means a word-combination Âńconcentrations of TECÂż: value, size, color? AC: We changed the words "concentrations of TEC" into "TEC values" in the text.

3. Page 5, a line 15: Table 1 has appeared earlier than the reference to it. It is desirable to specify number of station from Figure 1.

AC: We relocate Table 1 to be appeared after the reference to it. We also added the number of station to each of three stations discussed (CSMN, CREO, CNDE) in Table 1, the text and Fig. 4.

Original: To investigate the behavior of TEC related to the magnitude of the eclipse during the total solar eclipse on 9 March 2016, an analysis of two PRN numbers namely PRN 24 and PRN 12 from three GPS receiver stations was carried out. Those three stations are CNDE, CREO and CSMN (Figure 4). CSMN station is located closer to the totality path of the solar eclipse followed by CREO and CNDE stations. The location of GPS stations and trajectory of each PRN observed in this study are shown in Figure 4.

Revised: To investigate the behavior of TEC related to the magnitude of the eclipse during the total solar eclipse on 9 March 2016, an analysis of two PRN numbers namely PRN 24 and PRN 12 from three GPS receiver stations was carried out. Those three

stations are Station 26 (CSMN) in Sumenep, Madura Island, Station 30 (CREO) in Reo Ruteng, Sumbawa Island and Station 31 (CNDE) in Ende, Flores Island which can be seen from Fig.1 and Fig. 4. CSMN station is located closer to the totality path of the solar eclipse followed by CREO and CNDE stations. The location of GPS stations and trajectory of each PRN observed in this study are shown in Figure 4.

Some small remarks 1. Page 1, a line 13: to replace ÂńionoshpericÂż. AC: We have replaced "ionoshperic" with "ionospheric" 2. Page 1, a line 16, page 11, a line 7: faster to replace on less. AC: We have replaced "faster" with "less" 3. Page 3, a line 20: are to replace on is. AC: We have replace "are" with "is" 4. Page 8, a line 5: confuses a word ÂńfartherÂż. AC: We have replaced "farther" with "larger" 5. Page 10, a line 9: can be in a word-combination Âńthat that theÂż there is anything superfluous? AC: We have corrected "that that the" to "that the" 6. Page 11, a line 22: respon to replace on response. AC: We have corrected "respon" to "response" 7. Page 11, a line 28: respons to replace on response. AC: We have corrected "respons" to "response" 8. Page 11, a line 29: atmos to replace on Atmos. AC: We have corrected "atmos" to "Atmos" 9. Page 12, a line 10: to move the reference to page 12, a line 31. AC: We have moved the reference to its proper place 10. Page 12, a line 22: new to replace on New. AC: We have corrected "new" to "News" 11. Page 12, a line 29: march to replace on March. AC: We have corrected "march" to "March"

Please also note the supplement to this comment:
https://www.ann-geophys-discuss.net/angeo-2019-11/angeo-2019-11-AC1-supplement.pdf

**Supplement:**

[revised manuscript text omitted]
 Station 26 (CSMN) in Sumenep, Madura Island, Station 30 (CREO) in Reo Ruteng, Sumbawa Island and Station 31 (CNDE) in Ende, Flores Island which can be seen from Fig.1 and Fig. 4. CSMN station is located closer to the totality path of the solar eclipse followed by CREO and CNDE stations. The location of GPS stations and trajectory of each PRN observed in this study are shown in Figure 4.

25 The solar eclipse at CNDE began at 6:26 WIB and ended at 08:53 WIB, whereas at CREO it started at 06:25 WIB and ended at 08:51 WIB. Meanwhile at the CSMN the eclipse started at 06:21 WIB and ended at 08:41 WIB. The difference in the geographical position of the station causes a difference in the time of the eclipse, the magnitude of the obscuration and the magnitude of the eclipse in each station. Parameters of the total solar eclipse observed in all three stations are listed in the Table 1.

30

[Figure]

**Fig. 2.** (a), (c), (e), (g), (i) and (k) are VTEC distribution maps on 8 March 2018 at one hour interval started from 05:00:00 WIB (UTC + 7 hours) drawn in geographic coordinate system. (b), (d), (f), (h), (j) and (l) are the VTEC distribution maps at the same time interval on 9 March 2018 during the occurrence of the total solar eclipse.

[Figure]

**Fig. 3.** (a) Average VTEC over Indonesia on normal condition (8 March 2016) and during the total solar eclipse (9 March 2016). (b) The decrease of TEC value during the occurrence of total solar eclipse. (c) Average scintillation index S4during the solar eclipse.

**Table 1.** Local condition at the time of total solar eclipse at each stations

| Station No. | Longitude (E) | Latitude (S) | Time of contact (WIB) | | | Magnitude | Obscuration |
|---|---|---|---|---|---|---|---|
| | | | First contact | Max. contact | Last contact | | |
| 26 (CSMN) | 114.0596 | 7.7218 | 06:21 | 07:26 | 08:41 | 0.842 | 80.71 % |
| 30 (CREO) | 120.363 | 8.4743 | 06:25 | 07:33 | 07:51 | 0.792 | 74.54% |
| 31 (CNDE) | 121.6545 | 8.8595 | 06:26 | 07:35 | 08:53 | 0.773 | 72.14% |

The variability and decrease percentage of the TEC in PRN 24 and 12 during the solar eclipse on 9 March and during the normal condition one day earlier are plotted in Figure 5 (a), (b) and (c) for CNDE, CREO and CSMN stations, respectively. The time interval of measurements covers start time, maximum time when the solar eclipse reached its totality and end time as denoted by vertical S, M and E lines, respectively in the figure. The TEC decrease during the same recorded time for CNDE station measured on PRN 24 and PRN 12 is about 22.87% for PRN 24 and 29% for PRN 12. The TEC decrease on PRN 24 and PRN 12 for CREO station is respectively about 23.78% and 45.73%. Whereas the TEC decrease on PRN 24 and PRN 12 for CSMN station is 69.51% and 70.68%. From the above results, we see that the decline in TEC is directly proportional to the magnitude and the obscuration of the solar eclipse obscuration which indicates the amount of radiation received in the area.

The closer the PRN path is to the total eclipse path, the greater the decrease in the TEC is. This study also confirms previous research conducted by Sharma et al. (2010) discussing the ionospheric response when an annular solar eclipse occurred in India. The magnitude of TEC decrease as the distance between the observation station and the eclipse pathway is larger.

[Figure]

**Fig. 4.** Location of Station 26 (CSMN), Station 30 (CREO) and Station 31 (CNDE) and paths of the satellites labelled 
[revised manuscript text omitted]

---

## Referee Comment (RC2) · Anonymous Referee #2 · 17 Feb 2019

The work reports the decrease in VTEC over the Indonesian sector during the total solar eclipse on 09 March, 2016 based on data from 40 GPS stations distributed throughout the archipelago. The authors note that 1. VTEC recovery takes more time than the reduction during the eclipse and 2. The maximum reduction in VTEC is more as one goes closer to the path of totality. Is the inference 1 not known based on the work of one of the coauthor's earlier work (Muslims et al., 2016)? Although the authors acknowledge the possible contribution of plasma fountain process during the eclipse period, its role is not critically evaluated in the reduction of VTEC. The work has many

loose ends and the conclusions are not supported by necessary evidences.

A few specific comments are as follows.

1. Considering the occurrence of moderate storm on 06 March 2016, how do the authors rule out the contribution of negative ionospheric storm during the eclipse period? 2. Reduction in VTEC can also occur through plasma fountain effect essentially driven by the zonal electric field. How do the authors know that there is no electric field disturbance due to disturbance dynamo which can reduce VTEC over the archipelago through fountain effect? 3. How do the authors rule out prompt penetration and associated overshielding effects? It is known that overshielding process can generate westward electric field perturbations during daytime that can reduce VTEC over low latitudes through fountain effect. 4. The elevation angle cut-off of 10 degree will allow multi-path errors in the VTEC estimations. Elevation cut-off of at least 30 degree must be applied. 5. Figure 2: The reductions and increases in VTEC are not conspicuous. Quantitative descriptions are needed in the text. 6. Changes in S4 are negligible during the eclipse period. However, why the authors expect changes in S4 during local morning hours due to eclipse? 7. What do the authors mean when they state that the solar eclipse magnitude is greater than 1? 8. Is this work another repetition/extension of essentially the same work described in the Muslims et al. (2016) work? The same event is addressed in the Muslims et al. (2016) work which is evident from the title. 9. The authors need to put 1-sigma value to the average VTEC variation. If the decrease on 09 March, 2016 is within 1-sigma, then the authors cannot unambiguously attribute the reduction in VTEC to any physical process. 10. Please also note that the satellite and receiver biases together can account for the magnitude of reduction in VTEC during the eclipse (∼05 TECU). How do the authors remove these biases?

I am afraid that I am not able to accept this manuscript for publication in Annales Geophysicae.

[Figure]

2019.

---

## Referee Comment (RC3) · Anonymous Referee #3 · 23 Feb 2019

The work reports the decrease in VTEC over the Indonesian sector during the total solar eclipse on 09 March, 2016 based on data from 40 GPS stations distributed throughout the archipelago. The authors note that the time required of the VTEC archived maximum reduction since the initial contact is faster than the recovery phase during the eclipse period. and the possible contribution of plasma fountain process during the eclipse period, its role is not critically evaluated in the reduction of VTEC.

A few specific comments are as follows. 1. Considering the occurrence of moderate storm on 06 March 2016, The author describes the geomagnetic storm effect only on 7 March 2016 and therefore the authors take the VTEC data on 8 March 2016 to be the

reference to compare the eclipse effect. The storm initial phase occurs at 1700 UT on 6 March 2016 and reach the minimum at 2200 UT on 6 March 2016. However, the local time is later 7 to 11 hours for the UT time and the Dst index reaches the minimum value at daytime on 7 March 2016. Therefore, how do the authors rule out the contribution of negative ionospheric storm on 8 March 2016 and the data could be as the reference for comparison? 2. The same work described in the Muslims et al. (2016) work and the authors should describe what the different between two papers.

I am afraid that I am not able to accept this manuscript for publication in Annales Geophysicae.